# Phytochemical Characterization and In Vitro Anti-Inflammatory Evaluation in RAW 264.7 Cells of *Jatropha cordata* Bark Extracts

**DOI:** 10.3390/plants12030560

**Published:** 2023-01-26

**Authors:** Yazmín B. Jiménez-Nevárez, Miguel Angel Angulo-Escalante, Julio Montes-Avila, Araceli Guerrero-Alonso, Judith González Christen, Israel Hurtado-Díaz, J. Basilio Heredia, Eber Addí Quintana-Obregón, Laura Alvarez

**Affiliations:** 1Centro de Investigación en Alimentación y Desarrollo A.C. Carretera Eldorado km 5.5, Campo El Diez, Culiacán 80110, Mexico; 2Programa de Posgrado en Ciencias Biomédicas, Facultad de Ciencias Químico-Biológicas, Universidad Autónoma de Sinaloa, Ciudad Universitaria s/n, Culiacán 80010, Mexico; 3Centro de Investigaciones Químicas IICBA, Universidad Autónoma del Estado de Morelos, Avenida Universidad 1001, Col. Chamilpa, Cuernavaca 62209, Mexico; 4Laboratorio de Inmunidad Innata, Facultad de Farmacia, Universidad Autónoma del Estado de Morelos, Avenida Universidad 1001, Col. Chamilpa, C.P., Cuernavaca 62209, Mexico; 5Departamento de Madera Celulosa y Papel, Centro Universitario de Ciencias Exactas e Ingenierías, Universidad de Guadalajara, Km 15.5 Guadalajara-Nogales, Las Agujas, Zapopan 45100, Mexico; 6CONACYT-Centro de Investigación en Alimentación y Desarrollo A.C., Carretera Gustavo Enrique Astiazarán Rosas, No. 46, Col. La Victoria, Hermosillo 83304, Mexico

**Keywords:** *Jatropha cordata*, anti-inflammatory activity, NO production inhibition, terpenes, gas chromatography–mass spectrometry, NMR

## Abstract

The inflammatory process, although beneficial, can produce tissue damage and systemic damage when uncontrolled. Effective therapeutic alternatives with little or no side effects are of great therapeutic interest. This study aimed to determine the phytochemical composition of bark extracts from *J. cordata*, an endemic plant from México, and evaluate their in vitro anti-inflammatory activity. Hexane, ethyl acetate, and methanol extracts were characterized by qualitative phytochemical tests, and their bioactive groups were identified by ^1^H NMR and gas chromatography coupled to mass spectrometry (GC–MS). The extract’s anti-inflammatory activity was evaluated as nitric oxide (NO) production and their cytotoxicity by an MTS cell proliferation assay in lipopolysaccharide (LPS)-activated RAW 264.7 cells at concentrations of 1–100 μg/mL. The hexane extract contained fatty acids, fatty esters, phytosterols, alkanes, vitamin E, and terpenoids; the ethyl acetate extract showed fatty acids, fatty esters, aromatic aldehyde, phytosterols, vitamin E, and terpenoids, while the methanolic extract showed fatty esters, fatty acid, aromatics aldehydes, and alcohol. The ethyl acetate extract showed the highest inhibition of NO production, followed by the methanolic extract and the hexane extract, without affecting the viability of RAW 264.7 macrophage cells. The results suggest that *J. cordata* extracts are a potential source of bioactive compounds with anti-inflammatory potential.

## 1. Introduction

Chronic inflammation is a health problem that can lead to severe diseases. Using synthetic drugs for controlling chronic inflammation causes undesirable side effects, so their replacement by bioactive compounds of a natural origin is an important research activity. Species of the genus *Jatropha* have been used in the treatment of diabetes [1], and malaria [2], as well as for their anti-plasmodial [3], anti-leishmanial, anti-bacterial, anti-microbial [4], anti-molluscicidal [5], anti-carcinogenic, anti-proliferative [6], gastroprotective [7], and anti-inflammatory [8] activities. Of the 21 species of the *Jatropha* genus studied, 175 known bioactive compounds have been found, such as cyclic peptides, lignans [9], flavonoids [1], coumarins [10], alkaloids [11], monoterpenes [7], sesquiterpenes [3], diterpenes [3], and triterpenes [12]. 

*Jatropha cordata* (Ortega) Müll. Arg. is a plant widely distributed in northwestern México. The population uses an infusion obtained from the bark of its stem, applied topically or orally, to treat gastrointestinal problems and promote wound healing [13]. A chemical characterization study of flour and seed oil reported the presence of fatty acids and phorbol esters [14]. At the same time, Vega-Ruiz et al. [15] found phenolic acids and flavonoids with antioxidant activity in leaf flour and bark of *J. cordata*. The ethnomedical knowledge and phytochemical composition of *J. cordata* suggest that its bark could be an important source of bioactive compounds. Phytochemical profiles of hexane, ethyl acetate, and methanol extracts were determined by gas chromatography coupled to mass spectrometry (GC–MS) and nuclear magnetic resonance (NMR). In addition, nitric oxide inhibition was used to evaluate cytotoxic activity by the MTS cell proliferation method and in vitro anti-inflammatory activity was evaluated in RAW 264.7 macrophages. This work aimed to determine whether *J. cordata* is a natural source of bioactive compounds for treating chronic inflammation.

## 2. Results and Discussion

### 2.1. Yields

The average yields per extract from 100 g of *J. cordata* bark flour were: hexane (2.00%), ethyl acetate (1.52%), and methanol (4.40%).

### 2.2. Qualitative Analysis of Phytochemical Compounds

The qualitative phytochemical profile of the extracts revealed the presence of flavonoids, tannins, alkaloids, saponins, and terpenoids (Table 1). In the hexane extract, the Liebermann–Burchard and Salkowski reaction showed a color change to green, indicating the presence of triterpenes and/or steroids, chemical groups associated with the apolar hexane solvent. The above agrees with results obtained by [16], who described the presence of terpenoids, steroids, tannins, and glycosides in hexane extract obtained by the reflux method from *J. curcas* seed. In another phytochemical study performed on the hexane extract of bark and leaf of *J. cordata,* Vega-Ruiz et al. [15] reported the presence of sterols— coinciding with our results.

The ethyl acetate extract showed a moderate presence of tannins due to precipitate formation, of triterpenes and steroids by a color change to green and a slight presence of alkaloids due to slight turbidity and color change from red to orange, of saponins by foam formation, and of flavonoids by changing to blue coloration. The chemical groups identified correspond to those expected in the extraction with ethyl acetate solvent because it is a solvent of intermediate polarity. 

Methanol is a solvent of high polar affinity, which can extract alcohols, amines, esters, alkenes, halides, and alkanes. In our tests, all physicochemical tests were positive. Abundant precipitation was observed, indicating a high presence of tannins; a moderate color change to blue was observed, indicating the presence of flavonoids; low turbidity with a red to orange coloration was observed, indicating a low presence of alkaloids. Nazeema and Girija [17] reported the presence of alkaloids, flavonoids, and steroids in methanolic extracts obtained by the maceration of *J. gossippifolia* bark and of *J. curcas*. Similarly, Sharma et al. [16], analyzing methanolic extracts obtained by seed pressing and latex of *J. curcas*, reported the presence of alkaloids, saponins, phenolic compounds, and flavonoids. Serrano-Gallardo et al. [18], in the methanolic extract of aerial parts of *Jatropha dioca Sessé*, flavonoids and triterpenes were reported. Finally, in a methanolic extract of the bark and leaf of *J. cordata*, Vega-Ruiz et al. [15] showed the presence of flavonoids and tannins. 

### 2.3. GC–MS Analysis

Metabolic profiling of plant extracts refers to the analysis by the hyphenated technique GC–MS [19]. Accurate mass spectrometry and spectral data are processed with specific algorithms that provide a specific molecular formula, and then the metabolites are identified in available databases. Following this approach, 23 different metabolites were identified. The metabolic profiles of extracts from the bark of *J. cordata* included alcohols, alkanes, esters, fatty acids, and terpenoid compounds.

In the hexane extract, the following groups were found, in order of abundance: fatty esters (32.42%), fatty acids (31.89%), terpenoids (16.07%), vitamin E (4.17%), alkanes (3.4%), and phytol (0.49%). A total of 19 volatile compounds were found, of which 6 were present in percentages greater than 5%: 9,12-octadecadienoic acid (19.86%); 9,12,15-octadecatrienoic acid, methyl ester (13.91%); n-hexadecanoic acid (9.80%); γ-sitosterol (9.53%); hexadecanoic acid, methyl ester (6.58%); and 9,12-octadecadienoic acid, methyl ester (6.04%). The bioactive compounds´ retention times range from 18.57 to 35.88 min, which corresponds to the retention times for fatty acids. The fatty compounds´ abundance is directly related to their polarity and inversely related to their molecular weight (Table 2). Overall, the compounds identified in the hexane extract are common constituents of the cuticles and membranes of most plants [20]. 

In hexane extracts obtained by Soxhlet extraction of *J. gossypifolia* seed, Ogbobe and Akano [21] and Hosamani and Katagi [22] report the presence of fatty acids by GC–MS analysis. As for *J. cordata*, these authors showed the presence of palmitic and linoleic acids. In a study of hexane extracts of leaf, root, and bark of *J. curcas*, Chauhan et al. [23] identified alcohols, hydrocarbons, esters, ketones, and others by GC–MS, with insecticidal activity. In the hexane extract of *J. cordata*, fatty esters, alkanes, and phytosterols were identified. The three studies identified vitamin E and squalene.

On the other hand, Othman et al. [24] identified the presence of fatty acids, phenolic compounds, sterols, and terpenoids in hexane extracts obtained by the maceration of the leaves, fruits, bark, and root of *J. curcas.* These authors identified the presence of fatty esters, hexadecanoic acid methyl ester, and the fatty acids 9-octadecanoic acid methyl ester and octadecanoic acid, with anti-inflammatory potential in vitro. In the present study, the three mentioned compounds were also found in the hexane extract of *J. cordata* bark. The authors of [8], in hexane extracts obtained by maceration of the pulp, leaf, and seed of *J. platyphylla* and using the GC–MS technique, found a fatty acid, saturated alcohol, and alkane in the pulp; two sterols, a fatty acid, and an alcohol in the leaf; and three fatty acids, an aromatic acid and a terpene in the seeds. These authors reported γ-sitosterol in the leaf with a high anti-inflammatory potential in vitro, an activity attributed to the presence of sterols and terpenes. These results coincide with those obtained in the hexane extract of *J. cordata* bark.

Gámez-Meza et al. [14], studying hexane extracts in *J. cordata* seed flours, reported the presence of myristic, palmitic, palmitoleic, stearic, oleic, linoleic, linolenic, and arachidic acids. These results coincide with the bioactive compounds found in the present work, identifying in addition: a terpenoid (3,7,11,15-tetramethyl-2-hexadecen-1-ol), unsaturated fatty esters (9,12,15-octadecatrienoic acid, methyl ester; and the unsaturated fatty acid 9,12-octadecadienoic acid), and saturated fatty esters (docosanoic acid, methyl ester; and hexacosanoic acid, methyl ester) which have not been reported in the literature.

The ethyl acetate extract contained 12 volatile compounds. The predominant chemical groups found were aromatic aldehyde (benzaldehyde) with a retention time of 6.26 min, and an abundance of 45.34%; fatty acids (*n*-hexadecanoic acid (palmitic acid); 9,12-octadecadienoic acid; octadecanoic acid; with retention times of 19.08, 20.81, and 20.93 min and relative abundances of 13.22%, 20.08% and 2.86%, respectively), terpenoids (phytol, squalene, stigmasta-5,22-dien-3-ol, and γ-sitosterol with retention times of 17.69, 28.98, 34.80, and 35.87 min and relative abundances of 1.19, 1.00, 2.52, and 8.63%, respectively) (Table 2).

Othman et al. [24] reported the presence of fatty acids, phenolic compounds, sterol, and terpenoids in the methanolic extract of *J. curcas* root. These authors identified the presence of hexadecanoic acid methyl ester, 9-octadecanoic acid methyl ester, octadecanoic acid, terpene, and γ-sitosterol. The three mentioned compounds were also found in the ethyl acetate extract of *J. cordata* bark. In another study, with an ethyl acetate extract obtained by ultrasound from *J. curcas* seeds, Thi et al. [25] identified fatty acids—palmitic acid and α-tocopherol, and γ-sitosterol—chemical groups and compounds also found in *J. cordata* bark.

The methanolic extract contained 7 volatile compounds. The groups quantified were aromatic aldehyde (benzaldehyde and banzaldehyde dimethyl acetal with abundances of 54.46% and 1.53%, respectively) and fatty esters (6,9,12-octadecatrienoic acid methyl ester, palmitic acid methyl ester, and linoleic acid methyl ester, with abundances of 1.29%, 0.71%, and 0.52%, respectively (Table 2).

Rahman et al. [26] identified, by GC–MS, fatty acids and esters with antimicrobial activity in a methanolic extract obtained by maceration of *J. curcas* leaves. These authors reported the bioactive compounds: palmitic acid, methyl ester; linoleic acid, methyl ester; and palmitic acid—compounds also found in the methanolic extract of *J. cordata* bark. Furthermore, in a methanolic extract obtained by ultrasound from *J. curcas* seeds, Thi et al. [25] identified fatty acids. Alhaj et al. [27] identified fatty esters and fatty acids (palmitic acid) with anti-infertility activity in a methanolic extract from the fruit of *J. variegate*, which correspond to the chemical groups and bioactive compounds found in the methanolic extract of *J. cordata* bark.

On the other hand, phenolic compounds in a methanolic extract of *J. cordata* bark with antioxidant activity were reported by Vega-Ruiz et al. [15]. These authors identified, by HPLC, the following bioactive compounds: 3,4-dihydroxy-benzoic acid, 4-hydroxy-benzoic acid, caffeic acid, ellagic acid, gallic acid, gentisic acid, p-coumaric acid, syringic acid, sinapic acid, tannic acid, vanillic acid, apigenin, catechol, epicatechin, apigenin-7-O-rutinoside, rutin hydrate, isovitexin, pyrogallol, quercetin, rhoifolin, rutin, and vitexin. 

Finally, an unsaturated fatty ester (6,9,12-octadecatrienoic acid, methyl ester), two aromatic aldehydes (benzaldehyde; and benzaldehyde dimethyl acetal), and an alcohol (benzyl alcohol)—which had not been previously reported in the literature—were also identified in the methanolic extract of the present study. 

### 2.4. Nuclear Magnetic Resonance

The ^1^H NMR spectrum of the hexane extract indicated the presence of aliphatic hydrogens resembling those of fatty acids and terpenoids, according to the following evidence: (i) vinyl hydrogens in the region of δ 5.12–5.37; (ii) carbinol-type hydrogens in δ 3.67, (iii) allylic hydrogens in δ 2.75–2.84, (iv) α to carbonyl in δ 2.31, (iv) proton adjacent to double bond in δ 1.9–2.1, (vi) β carbonyl in δ 1.6, and (vii) signals assignable to methyls and methylenes in δ 0.88–1.40 (Figure 1a). According to Ravindranath et al. [28], the hexane extract of the aerial parts of *J. curcas* showed 20 compounds, from which 4 diterpenoids of the lathyran type and 2 compounds of the podocarpan type were identified by ^1^H NMR. 

In the ethyl acetate extract, the ^1^H NMR spectrum showed characteristic fatty acid ester and triterpene signals: (i) aromatic protons in the region δ 6.8–7.5, (ii) vinyl hydrogens in δ 5.10–5.40, (iii) methoxyls in δ 4.25, (4) carbinol-type hydrogens in δ 3.32, (v) allylic hydrogens in δ 2.76, (vi) α to carbonyl in δ 2.26, (vii) proton adjacent to a double bond in δ 1.9–2.1, (vi) proton β to a carbonyl in δ 1.6, and (ix) methyls and methylenes in δ 0.85–1.30 (Figure 1b). These results agree with Luo et al. [29], who isolated four tetracyclic triterpenes from the leaf and bark of *J. gossypifolia* from 95% EtOH, petroleum ether, and ethyl acetate.

The ^1^H NMR analysis of methanol extract showed characteristic signals for (i) aromatic hydrogens in δ 6.5–7.1, (ii) vinyl hydrogens in δ 5.2–5.35, (iii) carbinol hydrogens in δ 3.35–4.45, (iv) methoxyls in δ 3.27, (v) *α* to carbonyl in δ 2.10–2.9, (vi) proton adjacent to a double bond in δ 1.9–2.05, (vii) β to carbonyl in δ 1.52, and (viii) methyl and methylene groups in δ 0.84–1.30. These signals belong to fatty acids and aromatics (Figure 1c). Oskoueian et al. [30] report the presence of carboxyl, hydroxyl, methyl groups, fixed anionic, and cationic traces with antimicrobial, anti-inflammatory, and antioxidant activities in *J. curcas* flour.

### 2.5. In Vitro Anti-Inflammatory Activity

#### 2.5.1. Cell Viability

The cell viability of crude extracts in RAW 264.7 cells was determined at different concentrations (1, 10, 25, 50, and 100 μg/mL). The extracts and DMSO vehicle had no significant effect on decreasing cell viability at the concentrations tested compared to the LPS + Indo treatment (positive control). The viability of the extracts at all concentrations was superior to the LPS treatment (negative control), except for LPS + H 1 μg/mL (Figure 2).

An analysis of variance showed significant differences (*p* < 0.05) in the main effects of solvents and concentrations. Figure 3a shows that ethyl acetate and methanol presented similar viabilities and a lower cytotoxicity, while hexane showed a lower value and a higher cytotoxicity. A lower cytotoxicity effect was observed starting at 50 μg/mL.

The analysis of variance showed a significant (*p* < 0.05) interaction effect between the solvents used and concentrations (Figure 3b). Hexane and methanol extracts showed no overproduction (% viability > 125) of RAW 264.7 cells, while ethyl acetate, at a concentration of 100 μg/mL, showed slight overproduction, which may indicate an adverse treatment effect.

#### 2.5.2. Nitric Oxide Production (NO)

NO inhibition increased for the three extracts, with ethyl acetate and methanol showing the larger inhibition. At a concentration of 100 μg/mL, ethyl acetate presented more significant inhibition of nitric oxide (56.67%) than the LPS + Indo treatment (38.11%) (positive control) (Figure 4).

Ethyl acetate showed the highest nitric oxide inhibition, while the 50 and 100 μg/mL concentrations were the most effective (Figure 5a). The extracts showed a linear increasing percentage inhibition with concentration. Ethyl acetate obtained the highest percentage from the concentration of 25 μg/mL (Figure 5b).

The hexane extract of *J. cordata* bark did not present cytotoxicity at concentrations of 1–100 µg/mL (Figure 2). Likewise, given a higher presence of fatty esters and unsaturated fatty acids (Table 2), its anti-inflammatory activity, expressed as a percentage of NO inhibition, was lower than the activity of the commercial pharmacological control (Indo) for all concentrations, which could be attributed to the mechanisms of action developed for fatty acids. A review article by Rogero and Calder [31] reports that fatty acids stimulate an inflammatory response in vitro through TLR2 and TLR4. Furthermore, fatty acids induce COX-2 expression through an NFκB-dependent mechanism in a macrophage cell line. Diacylated and triacylated lipoproteins, peptidoglycans, and lipoteichoic acid are among the TLR2 receptor agonists. In addition, lauric acid induced inflammation through NF-κB when TLR2 was cotransfected with TLR1 or TLR6; however, this did not occur when TLR1, 2, 3, 5, 6, or 9 were transfected individually. Lauric acid, which does not solubilize in bovine serum albumin, induced activation of the NF-κB signaling pathway through TLR2, which dimerized with TLR1 or TLR6, and TLR4. Palmitate acid bound to TLR4 activates JNK and IKK-β protein kinases and increases the expression and secretion of pro-inflammatory cytokines.

The ethyl acetate extract showed no cytotoxicity at 1–100 µg/mL concentrations (Figure 2) due to the large presence of unsaturated fatty acids and terpenoids (Table 2). Its anti-inflammatory activity at the 100 µg/mL concentration, expressed as a percentage of NO inhibition, was higher than the activity of the commercial pharmacological control (Indo). This extract presented the highest anti-inflammatory potential, attributed to fatty acids, terpenoids, and aromatic compounds. The anti-inflammatory activity of this extract can be explained by the mechanisms of action developed for fatty acids and terpenes. Fatty acids inhibit the expression of inflammatory genes, such as COX-2, iNOS and IL-1, IL-2, and TNF-α in macrophages, and reduce the activation of the NF-κB pathway and the expression of cytokines and COX-2 induced by TLR agonists, such as lipopeptides (TLR2) and LPS (TLR4) in macrophages. Fatty acids present another mechanism to modulate the inflammatory response by binding to G protein-coupled receptor 120 (GPR120). Fatty acid-induced activation of GPR120 leads to the recruitment of β-arrestin 2 to the plasma membrane, where this protein binds to GPR120. Subsequently, the GPR120/β-arrestin 2 complex is internalized into the cytoplasmic compartment, where it binds to the TAK1-binding protein (TAB1). This process disrupts the association between TAB1 and growth factor β-activated kinase (TAK1), consequently reducing the activation of TAK1 and the activity of the IKKK-β/NF-κB and JNK/AP-1 signaling pathways, respectively. The TAB1/TAK1 junction is a point of convergence of stimuli induced by the TLR4 and TNF receptor (TNFR) signaling pathway. Mitigation of TAK-1 activation by DHA leads to a reduced expression of genes with pro-inflammatory actions, such as TNF-α and IL-6. 

Other mechanisms related to the effects of fatty acids relate to their ability to bind to peroxisome proliferator-activated receptors (PPARs), including PPAR-α, PPAR-**γ,** and PPAR- β/Δ isoforms. PPAR isoforms form heterodimers with the retinoid X receptor (RXR) and bind to peroxisome proliferator response elements (PPRE) in the region responsible for promoting target genes that are involved in lipid metabolism and the inflammatory response; subsequently, they modulate the expression of these genes [32]. PPAR-α and PPAR-**γ** activations reduce the expression of genes encoding proteins that exhibit pro-inflammatory actions by inhibiting NF-κB activation. Furthermore, the anti-inflammatory effects of fatty acids on this signaling pathway can be attributed to decreased nicotinamide adenine dinucleotide phosphate (NADPH) oxidase activity, leading to a reduced recruitment of TLR4 to lipid rafts and dimerization of TLR4. The decreased NADPH oxidase activity also decreases the production of reactive oxygen species, which, in turn, are necessary to activate the TLR4 signaling pathway. Another possible mechanism of action of fatty acids relates to the ability to incorporate DHA into the plasma membrane, which may lead to reduced TLR4 translocation for lipid raft formation. This decreases TLR4 pathway activation and consequently decreases NF-κB activation [31].

The anti-inflammatory effect of squalene triterpene occurs through inhibition of the NF-κB pathway and decreases the phosphorylated transcription factor P65-NF-κB, leading to a reduction in the pro-inflammatory response through the following steps: (i) Upon LPS stimulation that binds to the CD14/TLR4/MD2 receptor complex, the TLR4 receptor interacts with the MyD88 gene, which aggregates the signal and is subsequently transmitted to the interleukin receptor-associated kinase 1 (IRAK) (by phosphorylation). (ii) Phosphorylated IRAK causes activation of the transcription factor NF-κB through phosphorylation of the IκBα protein by the multiprotein IKK complex, which triggers its ubiquitination and proteasomal degradation. (iii) NFκB is then free to translocate to the nucleus to induce NF-κB target gene expression. (iv) Squalene decreases the phosphorylated P65-NF-κB transcription factor P65-NF-κB, which (v) ultimately leads to a reduction in pro-inflammatory responses [33,34]. The diterpene β-sitosterol has also been shown to exert an anti-inflammatory effect in LPS-stimulated BV2 cells (murine microglial cells) via inhibition of the p38, ERK, and NFk-B pathways [35]. The diterpene α-tocopherol inhibits the ADP enzyme, PKC and Akt signaling pathways, prostaglandin (PGE2), and COX production, inhibits 5-LOX-dependent TNF-α, upregulates PPARc protein mRNA, and increases PPARc transcriptional activity [36,37].

The cytotoxicity of the methanolic extract, at concentrations of 1–100 µg/mL, was measurably lower than the commercial pharmacological control (LPS + Indo) (Figure 2). Its anti-inflammatory activity at a concentration of 100 µg/mL expressed as a percentage of NO inhibition was higher than the commercial pharmacological control (LPS + Indo), which the presence of fatty esters can explain. Brejchova et al. [38] mentioned that anti-inflammatory effects are the second most important properties of PAHSAs. The compound 9-PAHSA blocks lipopolysaccharide-stimulated activation of dendritic cells and macrophages through the free fatty acid receptor 4 (FFAR4, GPR120) and modulates innate and adaptive immune responses in a mouse model of colitis and type 1 diabetes. The 5- and 9-PAHSAs have been used as standards to compare the anti-inflammatory effects of novel FAHFAs. Thus, PAHSAs have beneficial effects on metabolism and the immune system and could be used to target autoimmune, metabolic diseases such as type 1 diabetes. The first receptor identified for PAHSAs was FFAR4 (alternatively GPR120) which regulates GLUT4 translocation in adipocytes. Subsequently, FFAR1 (GPR40) and FFAR4 were identified as mediators of anti-inflammatory effects. Studies on liver and WAT crosstalk suggest that multiple G protein-coupled receptors (families) may be involved in signal transduction. Sialic acid-binding immunoglobulin-like lectins (Siglecs) are a family of cell-surface immune receptors on macrophages, monocytes, and neutrophils that bind sialic acid at terminal glycan residues. The compound 5-PAHSA binds Siglec5 and Siglec14 and suppresses interleukin-8 (IL-8) production in human monocytic cells.

## 3. Materials and Methods

### 3.1. Collection of Plant Material

*J. cordata* bark, with a diameter of 0.5 to 4 cm and length of 50 cm, was collected in March 2019 from plants cultivated in La Campana (Longitude (dec): −107.573889; Latitude (dec): 24.996667), located in the municipality of Culiacán, Sinaloa, México. The bark was transported in Ziploc plastic bags to the Bioresources Laboratory of CIAD in Culiacán. The bark was removed from the bark manually using a razor, washed with potable water, and dried in the shade at 25 °C for two weeks until reaching a humidity <10%. The dried material was ground (Thomas-Wiley mill, model 4) until passing through a #40 mesh. The *J. cordata* bark flour obtained was kept in hermetically sealed polyethylene bags and stored at 4 °C until use.

### 3.2. Extraction

Using Harborne [39] methodology, 100 g portions of ground dried bark were subjected to extraction by maceration using 500 mL in a mass-to-volume ratio of 1:5 of hexane, ethyl acetate, and methanol. Each mixture was homogenized for 24 h with an S-500 orbital shaker (VWR International, USA) at 3000 rpm, 20 °C, and under darkness. Hexane, ethyl acetate, and methanol had purities of 98.99%, 98.8%, and 99.9%, respectively (Baker, Phillipsburg, NJ, USA).

The crude extracts were filtered with a porcelain funnel and Whatman no. 1 paper, concentrated on a rotary evaporator at P↓ (365 mbar, 45 °C and 50 rpm, brand name BUCHI, Canada), and kept at 3 °C under darkness until use. Extractions for each solvent were performed in triplicate.

### 3.3. Qualitative Phytochemical Analysis

Qualitative phytochemical analysis of the crude extracts was performed in test tubes or by thin-layer chromatography (silica gel matrix with 254 nm fluorescent indicator, Sigma/Aldrich). The following reactions determined the characteristic qualitative phytochemical profile: (i) Salkowski and Lieberman-Burchard reaction for triterpenes and/or sterols; (ii) Dragendroff, Mayer, and Wagner test for alkaloids and saponins; (iii) Shinoda test for flavonoids; and (iv) reaction with 1.0% gelatin solution and quinine sulfate solution with FeCl_3_ for tannins [26,39,40]. The results were reported as the presence (+) or absence (-) of each group of phytochemical compounds in the crude extracts of *J. cordata*.

### 3.4. GC–MS Chromatographic Analysis

The volatile components present in the crude extracts were analyzed by GC–MS using an HP Agilent Technologies 6890 gas chromatograph equipped with an MSD 5973 quadrupole mass detector (HP Agilent) and an HP-5MS capillary column (length: 30 m; inner diameter: 0.25 mm; film thickness: 0.25 M). A constant flow of carrier helium was adjusted to the column at 1 mL per minute. The inlet temperature was set at 250 °C, while the oven temperature was initially maintained at 40 °C for 1 min and increased to 280 °C at 10 °C/min intervals. The mass spectrometer started in positive electron impact mode with an ionization energy of 70 eV. Detection was performed in selective ion monitoring (SIM) mode, and peaks were identified and quantified using target ions. Compounds were identified by comparing their mass spectra with the NIST library version 1.7a. Relative percentages were determined by integrating the peaks using GC ChemStation software, version C.00.01. The composition was reported as a percentage of the total peak area.

### 3.5. Nuclear Magnetic Resonance

All NMR spectra were recorded on a multinuclear VARIAN-MERCURY-200 instrument (Varian, Palo Alto, CA, USA) for ^1^H-NMR spectra in CDCl_3_ with tetramethylsilane (TMS) as an internal standard. Chemical shifts are reported in δ values.

### 3.6. In Vitro Anti-Inflammatory Activity 

In vitro anti-inflammatory evaluation of crude extracts of *J. cordata* was performed using a RAW 264.7 macrophage cell model. Even though the inflammatory process cannot be induced in vitro, similar conditions can be created. Stimulation of macrophages with LPS from *Escherichia coli* results in the activation of a series of signaling events that potentiate the production of inflammatory mediators. Then, the three crude extracts were applied as inhibitor treatments. The stages of the evaluation are detailed below.

#### 3.6.1. Cell Culture and Viability Assay

A murine macrophage cell line RAW 264.7 (Tib-71TM ATCC) was cultured in Dulbecco’s modified Eagle’s medium in a nutrient mixture F-12 (DMEM/F12 medium) supplemented with 10% heat-inactivated fetal bovine serum (FBS). Cells were maintained in a humidified atmosphere containing 5% CO_2_ at 37 °C and subcultured by scraping and seeding in 25 cm^2^ flasks. To assess cell viability, the cells (2 × 10^4^ cells/well in 200 μL of medium) were seeded in a 96-well plate and incubated for 24 h. Subsequently, cells were treated with crude *J. cordata* bark extracts at varying concentrations (1, 10, 25, 50, 50, 100 μg/mL) using DMSO as a vehicle (0.21%, *v/v*), indomethacin (30 μg/mL) as a positive control, and untreated cells as a negative control. After 2 h, inflammation was induced with lipopolysaccharide (LPS) at a concentration of 4 μg/mL (for wells with extracts, vehicle, indomethacin, and 100% stimulus control) as a pro-inflammatory stimulus and without LPS (negative control) incubating for 22 h. Cell viability was determined by 3-(4,5-dimethylthiazol-2-yl)-5-(3-carboxymethoxyphenyl)-2-(4-sulfophenyl)-2*H*-tetrazolium (MTS) assay by adding 20 μL of MTS solution to each well, before the cells were incubated for another 4 h. Optical density was measured at 490 nm in a microplate reader. Cell-free supernatants were collected and used for nitric oxide (NO) quantification. The following equation calculated percent cell viability (%*CV*):$$\%CV=\left(\frac{a_{S}}{{\bar{a}}_{LPS}}\right)\times 100$$
where *a_S_* = absorbance of the sample, and ${\bar{a}}_{\mathrm{LPS}}$ = average LPS absorbance.

#### 3.6.2. Nitric Oxide (NO) Production

The nitrite ion (NO^2-^), the stable final product of NO, an indicator of NO production in cell-free supernatants, was measured according to the Griess reaction. A volume of 50 μL of supernatant from each extract was mixed with 100 μL of Griess reagent (50 μL of 1% sulfanilamide and 50 μL of 0.1% N-(1-*n*aphthyl) ethylenediamine dihydrochloride in 2.5 % phosphoric acid) for 10 min, at room temperature. The optical density of the mixture, at 540 nm (OD540), was measured with a microplate reader, and the concentration of nitrite in the samples was calculated using a standard curve of *NaNO_2_* prepared in fresh culture medium [41,42].

The following steps determined the percentage inhibition of NO:(1)A calibration curve was determined using the concentrations 0, 1, 5, 10, 10, 20, 40, 60, 60, 100 µg/mL of *NaNO_2_*.
$$a=0.0075\times c_{NaNO_{2}}-0.0086$$
where *a* = absorbance and $c_{NaNO_{2}}=$ concentration of sodium nitrate.(2)The corrected absorbance, (*a*_*c*_), was calculated for each crude extract, at concentrations of 0, 1, 10, 25, 50, and 100 µg/mL by the difference:
$$a_{c}=a_{S}-{\bar{a}}_{NaNO_{2}}^{0}$$where *a*_S_ = absorbance of the sample and ${\bar{a}}_{NaNO_{2}}^{0}\;$= average absorbance at zero concentration of *NaNO**_2_* curve.(3)The concentration of *NaNO_2_* (µM) present in each of the extracts was determined by the equation:
$$c_{NaNO_{2}}^{\mu M}=\frac{a_{c}+c_{NaNO_{2}}}{a_{S}}$$where $c_{NaNO_{2}}^{\mu M}$= micromolar concentration of sodium nitrate.(4)The percentage of *NaNO_2_* in each extract was obtained by the following equation:
$$\%NaNO_{2}=\left(\frac{c_{NaNO_{2}}^{\mu M}}{LPS_{NaNO_{2}}^{\mu M}}\right)\times 100$$(5)Finally, the percentage inhibition of nitric oxide (%*I_NO_*) was calculated by the following equation:
$$\%I_{NO}=100-\%\;NaNO_{2}$$

### 3.7. Statistical Analysis

Viability and NO inhibition percentages were described by means and standard errors for each extract. One-way ANOVA analyzed the differences between extracts. Two-way ANOVA analyzed the effects of crude extracts and their concentrations. Comparisons between means were performed using Tukey’s test. *p*-values < 0.05 were considered statistically significant. The MINITAB 19 package was used.

## 4. Conclusions and Recommendations

The identification of chemical groups corresponds to the polarity of the solvents used. Hexane, an apolar solvent, extracted with large abundance, fatty ester, fatty acids, and terpenoids. Ethyl acetate, of intermediate polarity, extracted aromatic aldehyde, fatty acids, and terpenoids with large abundance. Methanol, a polar solvent, extracted aromatic aldehyde and fatty ester chemicals with higher abundance.

At the recommended concentration range (1 to 100 μg/mL) for RAW 264.7 macrophage cells, none of the extracts affected their viability. This means that none of the extracts are cytotoxic and it may be appropriate to evaluate biological properties of interest such as anti-inflammatory, anti-cancer, and antioxidant activities.

Regarding their anti-inflammatory activity, ethyl acetate and methanol extracts at the concentration of 100 μg/mL showed greater NO inhibition (anti-inflammatory activity) than the pharmacological control (indomethacin). Thus, both extracts are potential sources of anti-inflammatory products.

Aromatic aldehyde, found in the ethyl acetate and methanol extracts in significant abundance, has been reported to possess an anti-inflammatory activity in other plants, which explains the high anti-inflammatory activity of these extracts.

In order to enhance our knowledge about *J. cordata* potential, we recommend that future research focus on obtaining and purifying fractions of the ethyl acetate extract. This will allow the characterization of the phytochemical groups and compounds exerting the anti-inflammatory activity.

## Figures and Tables

**Figure 1 plants-12-00560-f001:**
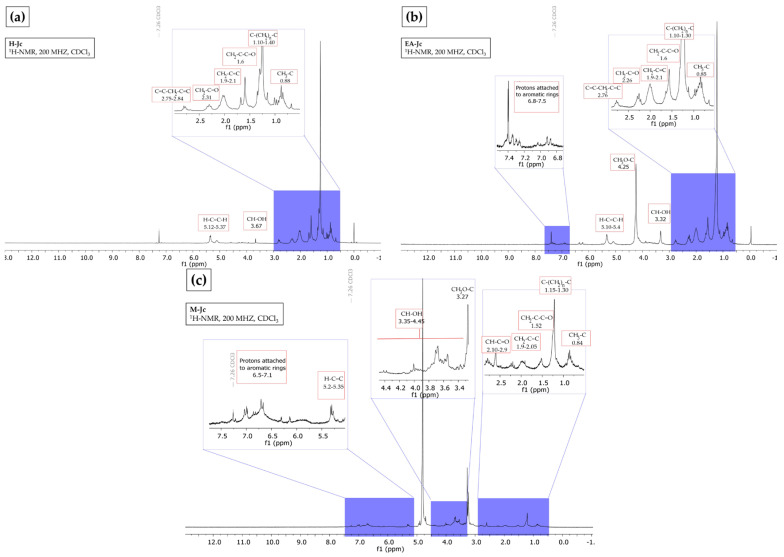
Hydrogen nuclear magnetic resonance (^1^H NMR) spectrum (CDCl_3_, 200 MHz). (**a**) Hexane from *J. cordata* bark extract, fatty acids and terpenes profile; (**b**) ethyl acetate from *J. cordata* bark extract, fatty acids and terpenes profile; (**c**) of methanol from *J. cordata* bark extract, fatty acids and aromatics profile.

**Figure 2 plants-12-00560-f002:**
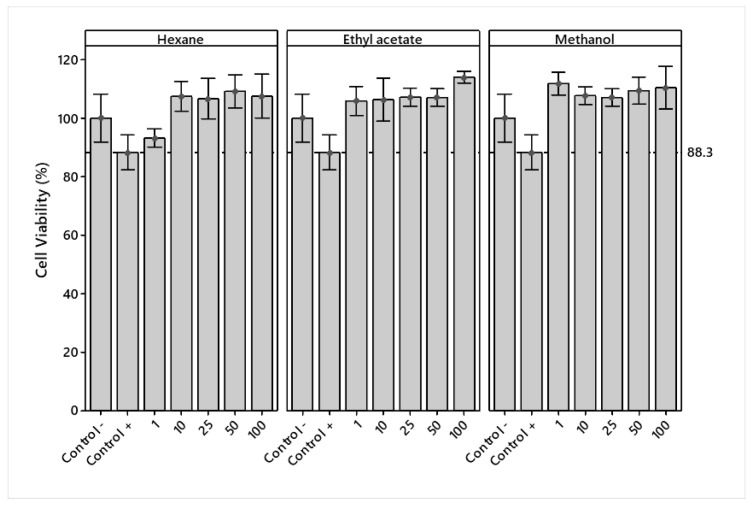
Mean percentage ± standard deviation of viability in LPS-stimulated RAW 264.7 macrophages treated with hexane, ethyl acetate, and methanol extracts of *Jatropha cordata* bark.

**Figure 3 plants-12-00560-f003:**
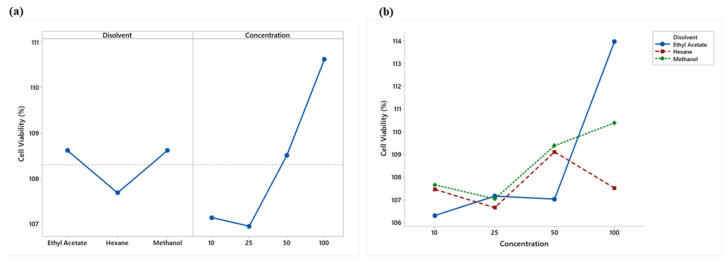
(**a**) Means plot of solvent and concentration means; (**b**) Means plot of the extract-concentration interaction of LPS-stimulated RAW 264.7 macrophages.

**Figure 4 plants-12-00560-f004:**
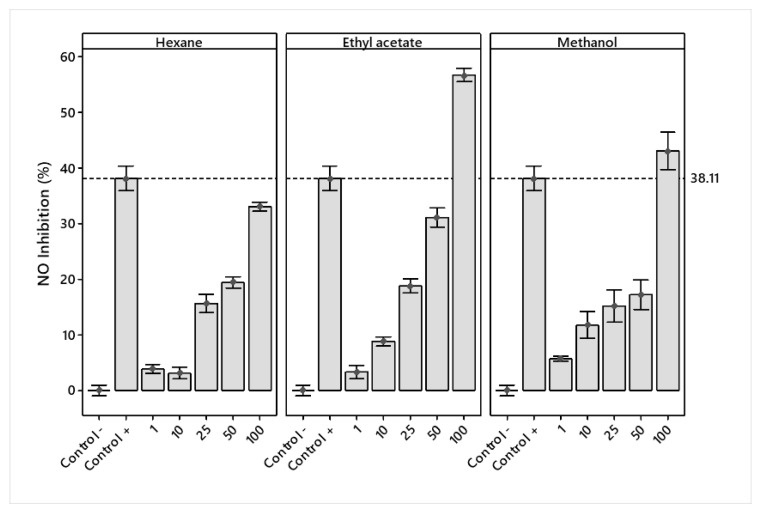
Mean percentage ± standard deviation of LPS-stimulated nitric oxide inhibition in RAW 264.7 macrophages treated with hexane, ethyl acetate, and methanol extracts of *J. cordata* bark.

**Figure 5 plants-12-00560-f005:**
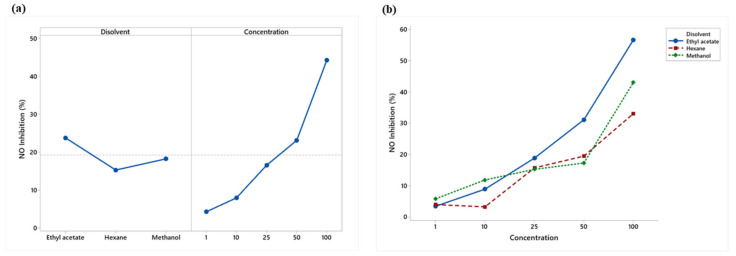
(**a**) Means plot of solvent and concentration means; (**b**) Means plot of extract-concentration interaction of RAW 264.7 macrophages stimulated with LPS.

**Table 1 plants-12-00560-t001:** Qualitative analysis of phytochemical compounds from the bark of *J. cordata*.

Phytochemical constituents	H-Jc	EA-Jc	M-Jc
Tannins	-	++	+++
Alkaloids	-	+	+
Saponins	-	+	+
Flavonoids	-	+	++
Triterpenes or steroids	+++	++	+

(-): No turbidity or precipitation; (+): Slight cloudiness or precipitation; (++): Medium turbidity or precipitation; (+++): High turbidity or precipitation. Solvents; H-Jc: hexane, EA-Jc: ethyl acetate, and M-Jc: methanol.

**Table 2 plants-12-00560-t002:** GC–MS analysis of bark extracts of *J. cordata*.

Phytochemical Group		Identified Compound	Chemical Formula	Molecular Weight	RT (min)	Abundance (%)		
						H-Jc	EA-Jc	M-Jc
Aromatic Aldehyde		Benzaldehyde	C_7_H_6_O	106	6.26	ND	45.34	54.46
		Benzaldehyde dimethyl acetal	C_9_H_12_O_2_	152	8.35	ND	ND	1.53
Alcohol		Benzyl alcohol	C_7_H_8_O	108	7.58	ND	ND	1.91
Terpenoids		3,7,11,15-tetramethyl-2-hexadecen-1-ol (phytol)	C_20_H_40_O	296	17.69	0.49	1.19	ND
		Squalene	C_30_H_50_	410	28.98	1.72	1.00	ND
		Stigmasta-5,22-dien-3-ol	C_29_H_48_O	412	34.80	4.33	2.52	ND
		γ-sitosterol	C_29_H_50_O	414	35.87	9.53	8.63	ND
Fatty acid	Saturated	*n*-hexadecanoic acid (palmitic acid)	C_16_H_32_O_2_	256	19.08	9.80	13.22	0.48
	Unsaturated	9,12-octadecadienoic acid	C_18_H_32_O_2_	280	20.81	19.86	20.08	ND
	Saturated	octadecanoic acid	C_18_H_36_O_2_	284	20.93	2.23	2.86	ND
Fatty ester	Saturated	Hexadecanoic acid, methyl ester (palmitic acid methyl ester)	C_17_H_34_O_2_	270	18.57	6.58	0.71	0.71
	Unsaturated	9,12-octadecadienoic acid, methyl ester (linoleic acid methyl ester)	C_19_H_34_O_2_	294	20.21	6.04	0.46	0.52
	Unsaturated	6,9,12-octadecatrienoic acid, methyl ester	C_19_H_32_O_2_	292	20.27	ND	1.04	1.29
	Unsaturated	9,12,15-octadecatrienoic acid, methyl ester	C_19_H_32_O_2_	292	20.29	13.91	ND	ND
	Saturated	octadecanoic acid, methyl ester	C_19_H_38_O_2_	298	20.48	1.39	ND	ND
	Saturated	Eicosanoic acid, methyl ester	C_21_H_42_O_2_	326	22.24	0.39	ND	ND
	Saturated	Docosanoic acid, methyl ester	C_23_H_46_O_2_	354	24.34	1.06	ND	ND
	Saturated	Tetracosanoic acid, methyl ester	C_25_H_50_O_2_	382	27.50	1.15	ND	ND
	Saturated	Hexacosanoic acid, methyl ester	C_27_H_54_O_2_	410	30.26	0.74	ND	ND
	Saturated	Octacosanoic acid, methyl ester	C_29_H_58_O_2_	438	32.68	1.17	ND	ND
Alkane		Nonacosane	C_29_H_60_	408	29.85	2.10	ND	ND
		Heptacosane	C_27_H_56_	380	32.19	1.30	ND	ND
Vitamin E		α-tocopherol	C_29_H_50_O_2_	430	32.79	4.17	2.96	ND
Grand Total						87.96	100.0	60.9

H-Jc: hexane extract; EA-Jc: ethyl acetate extract; M-Jc: methanol extract; ND: Not detected.

## Data Availability

Not applicable.

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
