# Peer review of "Phytochemical Characterization and In Vitro Anti-Inflammatory Evaluation in RAW 264.7 Cells of Jatropha cordata Bark Extracts"

_plants, 2023, doi:10.3390/plants12030560_

Round 1
Reviewer 1 Report
The inflammatory process, although beneficial, uncontrolled can produce tissue damage and systemic damage. Effective therapeutic alternatives with little or no side effects are of great therapeutic interest.
The abstract should be refined, so that it has fluency.
The study is carried out according to a well-established protocol, using high-performance methods and equipment that allow obtaining reliable results.
The description of the methods should also be revised. The inflammatory process cannot be induced in vitro, but similar conditions can be created. Stimulation of macrophages with LPS results in the activation of a series of signaling events that potentiate the production of inflammatory mediators.
The conclusions should contain a sentence highlighting the importance of this study and the results obtained.
I didn't understand the recommendations.
Author Response
Answers to the reviewer´s comments
- Reviewer comment. The inflammatory process, although beneficial, uncontrolled can produce tissuedamage and systemic damage. Effective therapeutic alternatives with little or no side effects are of great therapeutic interest.
Answer. The comment was incorporated at the abstract.
- Reviewer comment. The abstract should be refined, so that it has fluency.
Answer. The abstract was rewritten so its reading flows easily.
- Reviewer comment. The study is carried out according to a well-established protocol, using high-performance methods and equipment that allow obtaining reliable results.
- Reviewer comment. The description of the methods should also be revised. The inflammatory process cannot be induced in vitro, but similar conditions can be created. Stimulation of macrophages with LPS results in the activation of a series of signaling events that potentiate the production of inflammatory mediators.
Answer. The following paragraph shows how we attended this comment.
In vitro anti-inflammatory evaluation of crude extracts of J. cordata was performed using a RAW 264.7 macrophage cell model. Even though, the inflammatory process cannot be induced in vitro, similar conditions can be created. Stimulation of macrophages with LPS from Escherichia coli results in the activation of a series of signaling events that potentiate the production of inflammatory mediators, then the three crude extracts were applied as inhibitor treatments. The stages of the evaluation are detailed below.
- Reviewer comment. The conclusions should contain a sentence highlighting the importance of this study and the results obtained.
Answer. The following conclusion was added:
The study showed that ethyl acetate extract, at concentrations greater than 50 μg/mL, does not affect viability in RAW 264.7 cells and presents the highest NO inhibition. Therefore, J. cordata is a potential source of chemical compounds capable of inhibiting iNO-induced inflammation.
- Reviewer comment. I didn't understand the recommendations.
Answer. The following recommendations was added:
In order to enhance our knowledge about J. cordata potential, we recommend to obtain and purify fractions of the ethyl acetate extract. This will allows us to know the phytochemical groups and the compounds exerting the anti-inflammatory activity.

Reviewer 2 Report
Throughout the text: the names of the compounds should be written in small letter (eg. line 107-109, 134-136, 141-145, etc.)
Line 103 Based on the discussion, the statement that J. curacs represent the plant with most chemical and pharmacological studies is not clear.
Generally, Tables 2, 3 and 4 should be merged, as all represent GC-MS analysis.
Table 2. Please check abundance and RTs, seems to be switched.
Line 150, 173, etc. Do not start with citation: “[24] reported the presence of fatty acids,…” – “It was reported previously,…”
Table 3 – phytol Is terpene; also, why are fatty esters in fatty acid part, and at the end of the table fatty ester?
Figure 1. – Should be enlarged as relevant data are not observable. In addition, it should be given in full scale (up to 12 ppm), as carboxylic hydrogens are expected to be seen due to high percentages of acids.
Line 248-250: Please clarify how methanol was not significant (17.20%) when compared to hexane (19.41%)? With showed deviation, seems to be similar?
Conclusion and recommendations should be integrated under one chapter (Conclusion) and without 4.1., 4.2., etc.
Author Response
Answer to the reviewer´s comments
- Reviewer comment. Throughout the text: the names of the compounds should be written in small letter (eg. line 107-109, 134-136, 141-145, etc.)
Answer. This recommendation was atended in the text at tha corresponding numbered lines.
- Reviewer comment. Line 103 Based on the discussion, the statement that J. curacs represent the plant with most chemical and pharmacological studies is not clear.
Answer. The sentence was eliminated
- Reviewer comment. Generally, Tables 2, 3 and 4 should be merged, as all represent GC-MS analysis.
Answer. This recommendation was attended at the text.
- Reviewer comment. Table 2. Please check abundance and RTs, seems to be switched.
Answer. This recommendation was attended at the text.
- Reviewer comment. Line 150, 173, etc. Do not start with citation: “[24] reported the presence of fatty acids,…” – “It was reported previously,…”
Answer. This recommendation was attended at the text.
- Reviewer comment. Table 3 – phytol Is terpene; also, why are fatty esters in fatty acid part, and at the end of the table fatty ester?
Answer. The compounds were classified accordingly to their phytochemical group.
- Reviewer comment. Figure 1. – Should be enlarged as relevant data are not observable. In addition, it should be given in full scale (up to 12 ppm), as carboxylic hydrogens are expected to be seen due to high percentages of acids.
Answer. The NMR spectra were enlarged to a full scale of 12 ppm. However, the carboxylic hydrogens remain unobservable because they are interchangable protons.
- Reviewer comment. Line 248-250: Please clarify how methanol was not significant (17.20%) when compared to hexane (19.41%)? With showed deviation, seems to be similar?
Answer. The statement was eliminated from the text, since at the concentration of 50 μg/mL any extract was higher than the positive control.
- Reviewer comment. Conclusion and recommendations should be integrated under one chapter (Conclusion) and without 4.1., 4.2., etc.
Answer. This recommendations was atended in the text.

Round 2
Reviewer 2 Report
Table 2:
* 3,7,11,15-tetramethyl-2-hexadecen-1-ol (phytol)
* please add the sum of identified compounds
* Two phytosterols: are they not terpenoids from the biosynthetic pathway?
4. Conclusion: Please statements 4.1.-4.6. do in the form of text.
Author Response
Answer to the reviewer´s comments
- Reviewer comment. 3,7,11,15-tetramethyl-2-hexadecen-1-ol (phytol).
Answer. This recommendation was attended at the text.
- Reviewer comment. Please add the sum of identified compounds.
Answer. This recommendation was attended at the text.
- Reviewer comment. Two phytosterols: are they not terpenoids from the biosynthetic pathway?
Answer. This recommendation was attended at the text.
- Reviewer comment. Conclusion: Please statements 4.1.-4.6. do in the form of text.
Answer. The conclusions have been enhanced to include the major findings of this research.
